# Diagnostic Challenge in Rapidly Growing Langerhans Cell Histiocytosis with Aneurysmal Bone Cyst in the Maxilla: A Case Report

**DOI:** 10.3390/diagnostics12020400

**Published:** 2022-02-03

**Authors:** Ye Rin Hwang, Kyung Mi Lee, Hyug-Gi Kim, Kiyong Na

**Affiliations:** 1Department of Radiology, Kyung Hee University College of Medicine, Kyung Hee University Hospital, #23 Kyunghee-daero, Dongdaemun-gu, Seoul 02447, Korea; curryking227@daum.net; 2Department of Pathology, Kyung Hee University College of Medicine, Kyung Hee University Hospital, #23 Kyunghee-daero, Dongdaemun-gu, Seoul 02447, Korea; raripapa@gmail.com

**Keywords:** Langerhans cell histiocytosis, pediatric malignant tumor, aneurysmal bone cyst

## Abstract

Langerhans cell histiocytosis (LCH) is a rare neoplastic disorder characterized by the clonal proliferation of CD1a +/CD 207 + dendritic cells, whose features are similar to those of epidermal Langerhans cells. LCH is more common in children than in adults. Localized osteolytic lesions in the craniofacial bones are the most common manifestations of LCH. However, LCH can also present as a multifocal and multisystem disease with poor prognosis. Locally aggressive LCH needs to be differentiated from various diseases such as osteomyelitis, malignant bone tumors, and soft tissue sarcomas. However, it is difficult to diagnose, since the imaging findings are nonspecific. We report a case of a highly aggressive LCH in the maxilla accompanied by a fluid-fluid level.

## 1. Introduction

Langerhans cell histiocytosis (LCH) is a myeloid dendritic cell disorder that is more common in children than adults [1,2,3,4,5]. A localized bone lesion in craniofacial bone is most common, but LCH also appears in a multifocal multisystem form with a poor prog-nosis [1,2,3,4,5]. Locally aggressive LCH needs to be differentiated from various diseases such as osteomyelitis, malignant bone tumor, and soft tissue sarcoma but it is difficult to diag-nose because the imaging findings are nonspecific. We would like to introduce the LCH accompanied by a fluid-fluid level that occurred in the maxilla.


## 2. Case Presentation

A 27-month-old girl was admitted to our institution with right periorbital swelling without any history of trauma. The patient initially visited a local pediatrician, and conservative management was performed with an ophthalmic ointment and eye drops for the swelling of the right eyelid for two weeks. The swelling did not improve after the treatment, and skin redness developed. The pediatrician suspected periorbital cellulitis and started antibiotic treatment with cefaclor, but there was no improvement. Thus, the patient was transferred to our hospital. She had no medical history, and vaccination had been performed according to the normal schedule. The laboratory findings at the first visit revealed a C-reactive protein level of 0.59 mg/dL, and the erythrocyte sedimentation rate was slightly elevated to 57 mm/h. However, no other abnormal findings were noted.

The initial orbital computed tomography (CT) scan revealed a destructive lesion centered on the anterior wall of the right maxillary sinus. The lesion had formed a large soft tissue mass protruding into the maxillary sinus with expansile growth. A small area with fluid attenuation was observed in the anterior portion of the mass (Figure 1A). On initial T1-weighted magnetic resonance imaging (MRI), most of the mass showed an isointense signal when compared with the surrounding muscles (Figure 1B). T2-weighted imaging showed a heterogeneously high signal intensity. The fluid-attenuated focus observed in the CT scan was present in the cystic portion and accompanied by a fluid-fluid level, leading to the suspicion of an aneurysmal bone cyst (Figure 1C). The mass showed a heterogeneous enhancement after intravenous injection of a contrast medium (Figure 1D). Additionally, the mass showed the invasion of the ipsilateral inferior orbital wall and extraconal space. Soft tissue edema and fat infiltration were observed in the adjacent right cheek and periorbital space (Figure 1E). The maxillary alveolar processes and maxillary teeth were intact. The mass was characterized by rapid growth within two weeks.

The differential diagnosis depends on the clinical presentation and radiologic findings. Due to the large size and infiltrative nature of the soft tissue mass with a very rapid growth pattern, we considered the possibility of a soft tissue sarcoma such as rhabdomyosarcoma. We also considered the possibility of a malignant primary bone tumor accompanied by a secondary aneurysmal cyst, since there was a small cystic portion with a fluid-fluid level within the mass. However, the lesion was confirmed to be LCH through surgical biopsy (Figure 2A). The clusters of histiocytic cells show reactivity for CD1a and S-100 and were negative for H3, 3G34W (Figure 2B). A BRAF V600E mutation was not detected. Subsequently, the patient underwent several skeletal radiography, bone scintigraphy, and positron emission tomography-CT examinations to check for involvement of other organs. However, no lesions were observed except for that in the maxilla. The patient immediately received systemic therapy with vinblastine and prednisolone. The follow-up MRI of the orbit after two months revealed that the size of the lesion had decreased and that the accompanying periorbital soft tissue infiltration and maxillary sinusitis had improved (Figure 1F).

## 3. Discussion

LCH is a myeloid dendritic cell disorder in which abnormal immune cells present the same antigens (CD1a, S100, and CD207) in Langerhans cells in the skin and mucosa and infiltrate into various organs. The disease is more common in children, with a reported incidence of 0.2–2.0 cases per 100,000 children under 15 years of age [1,2,3,4,5,6]. D’Ambrosio reported that the average age of LCH patients was four years and ranged between four months and 24 years [6]. The highest incidence rate is observed among infants less than one year old [7]. LCH can involve the bone marrow, central nervous system, skin, lymph nodes, lung, liver, and spleen [1,2,3,4,5]. Focal bone lesions are the most common radiographic manifestations in LCH. In the axial skeleton, the skull is the most commonly involved site [1,2,3,4]. In cases of craniofacial involvement, mandibular lesions are more common in adults, while orbital lesions are more common in children [3,8]. The reported incidence of orbital and periorbital involvement in LCH is 12–20% [9,10], which has increased in frequency in the literature over the past years [11]. Patients typically present with proptosis and facial swelling.

Typical radiographic findings of LCH involving the skeleton include well-defined punched-out osteolytic lesions that can coalesce with each other with an increase in their size. Enhanced soft tissue lesions in the osteolytic area are observed on CT and MRI [1,2,3,4,5]. LCH is relatively common among non-odontogenic tumors of the facial bones in children [3].

It is difficult to diagnose LCH when the imaging findings are more complex. If sequestrum is found in the osteolytic lesions, osteomyelitis should be considered as a differential diagnosis. Pediatric patients with LCH and bone involvement often have adjacent soft tissue involvement. Extensive tumor infiltration into the adjacent tissues may result in swelling and pain [1,2,3,4,5]. Facial swelling is a common clinical problem in the pediatric LCH population. The origins of a facial swelling or mass can vary from congenital causes to acquired conditions such as infection and benign to malignant masses. In other words, aggressive and heterogeneous imaging features are nonspecific and can also be observed in soft tissue sarcomas and other malignant bone tumors. In patients with primary orbital and periorbital involvement, osteomyelitis, metastatic neuroblastoma, Ewing sarcoma, rhabdomyosarcoma, chloroma and lymphoma must be excluded. Among soft tissue sarcomas, rhabdomyosarcoma is the most common tumor in children and is most commonly observed in the head and neck region [12]. Osteosarcoma is the most common malignant primary bone tumor [12], but it is most commonly observed in the long bones, and craniofacial bones are rarely involved. Ewing sarcoma is the second most common malignant primary bone tumor in children, and it is common in both long and flat bones [12]. Metastatic neuroblastoma is characteristically involved in the posterolateral part of the orbit, frontal bone and greater wing of the sphenoid bone [13]. In cases of nonspecific findings, a histopathological confirmation is essential for a definitive diagnosis.

In the present case, LCH was difficult to diagnose due to the fluid-fluid level inside the mass. Due to the presence of a fluid-fluid level in an osteolytic mass, the possibility of a primary malignant osteogenic tumor with a secondary aneurysmal bone cyst as well as that of a soft tissue sarcoma that grows aggressively into the surrounding tissues could not be ruled out. Several case reports have described the presence of fluid-fluid (or hemorrhage-fluid) levels in LCH [5,8,14,15,16]. Nabavizadeh et al. [16] reported a series of 11 pediatric cases of skull lesions with fluid-fluid levels. Among these, four lesions were LCH, three were aneurysmal bone cysts, three were cephalohematomas, and one was metastatic neuroblastoma [16]. The fluid-fluid level is a nonspecific feature that can appear not only in benign lesions such as simple bone cysts, primary aneurysmal bone cysts, and fibrous dysplasia, but also in malignant bone tumors such as osteosarcoma and giant cell tumors. Hence, it is important to consider its possibility in LCH.

BRAF is considered one of the most common and well-known mutated kinases in the human center, which drives mutations in several tumors including colorectal cancer, non-small cell lung cancer, thyroid cancer and melanoma [17,18]. Its V600E mutation accounts for more than 90% of BRAF-activating mutations [19]. Previous studies have shown that BRAF V600E mutations are present in an average of 47% of LCH cases (from 0 to 64%) [20]. Feng et al. reported that patients from East Asia showed a lower frequency than other ethnic groups, regardless of age [18]. BRAF V600E mutation was not detected in our case. BRAF mutation in LCH cases significantly elevates phospho-extracellular signal-regulated kinase expression, which suggests the activation of the mitogen-activated protein kinase pathway, suppressing cell migration and augmenting cell survival [21]. The clinical significance of BRAF V600E in LCH is still contradictory because of a low incidence and high misdiagnosis rate.

## 4. Conclusions

The findings from the present report suggest that in cases of a rapidly growing bone tumor with or without an invasive pattern or aneurysmal bone cyst on imaging findings, LCH should be suspected based on the patient’s age and imaging findings. BRAF V600E mutation is helpful for diagnosis and for treatment in LCH patients, and we should remember that ethnic background appears to influence the risk of LCH development.

## Figures and Tables

**Figure 1 diagnostics-12-00400-f001:**
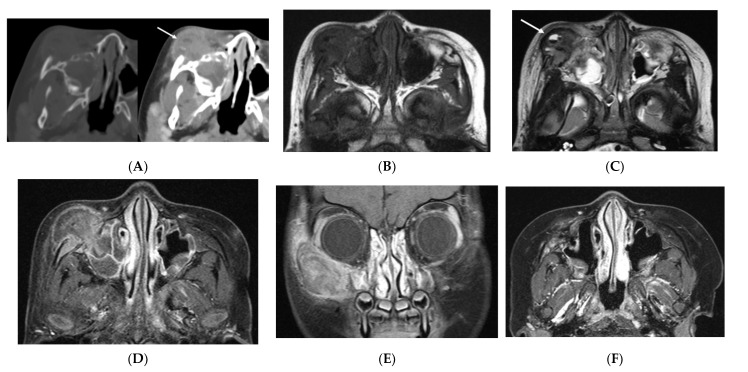
27-month-old girl with rapidly progressive facial swelling and redness. (**A**) Axial orbital CT bone and soft tissue window images show an expansile osteolytic lesion with associated soft-tissue mass involving the anterior wall of the right maxillary sinus and periorbital area. On axial and coronal MR imaging, the mass shows an (**B**) iso signal intensity (SI) on T1WI and (**C**) mixed high and low SI on T2WI, (**D**,**E**) with heterogenous enhancement. A small fluid-fluid level in the anterior aspect of the mass is observed (arrow). Direct invasion of the right inferior orbital wall and adjacent extraconal space is also observed. (**F**) On follow-up MR imaging after two months, the size and enhancement of the mass has decreased, and peritumoral infiltration and maxillary sinusitis are also improved. CT, computed tomography; MR, magnetic resonance; T1WI, T1-weighted imaging; T2WI, T2-weighted imaging.

**Figure 2 diagnostics-12-00400-f002:**
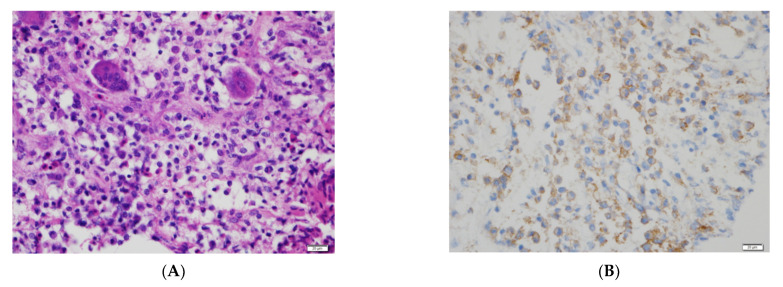
Biopsy results from the right periorbital mass. (**A**) Abundant Langerhans cells are seen, with scattered eosinophils and multinucleated giant cells (hematoxylin and eosin). (**B**) CD1 immunostaining demonstrates a membranous staining pattern with perinuclear dots.

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
