# Peer review of "Diagnostic Challenge in Rapidly Growing Langerhans Cell Histiocytosis with Aneurysmal Bone Cyst in the Maxilla: A Case Report"

_diagnostics, 2022, doi:10.3390/diagnostics12020400_

Round 1
Reviewer 1 Report
The authors present a rare case of a complicated course of bone lesion in the course of Langerhans cell histiocytosis.
The presentation is clear and informative.
Remarks:
I think that some information regarding frequency of such lesions as presented, should be added
Bibliography should be verified according editor guidelines.
Author Response
Reviewer A's comments #1> I think that some information regarding frequency of such lesions as presented, should be added
Reply: Thank you for your comments. According to your suggestion, we added the frequency of LCH in the discussion session as follow:
LCH is a myeloid dendritic cell disorder in which abnormal immune cells present the same antigens (CD1a, S100, and CD207) in Langerhans cells in the skin and mucosa and infiltrate into various organs. The disease is more common in children with a reported incidence of 0.2-2.0 cases per 100,000 children under 15 years of age [1-6]. D’ Ambrosio has been reported that the average age of LCH patients was 4 years and ranged between 4 months and 24 years [6]. LCH can involve the bone marrow, central nervous system, skin, lymph nodes, lung, liver, and spleen [1-5]. Focal bone lesions are the most common radiographic manifestations in LCH. In the axial skeleton, the skull is the most commonly involved site [1-4]. In case of craniofacial involvement, mandibular lesions are more common in adults, while orbital lesions are more common in children [3, 7]. The reported incidence of orbital and periorbital involvement in LCH is 12-20% [8, 9] which has increased in frequency in the literature over the past years [10]. Patients typically present with proptosis and facial swelling.
Reviewer A's comments #2> Bibliography should be verified according editor guidelines.
Reply: Thank you for your comments. According to your suggestion, we’ve modified the bibliography.
Reviewer 2 Report
Comments to the Corresponding Author
In the paper titled: "Diagnostic Challenge in Rapidly Growing Langerhans Cell Histiocytosis with Aneurysmal Bone Cyst in the Maxilla: A Case Report", the aim was to evaluate numerous differential diagnostic possibilities, according to the radiological, CT and MRI features.
The paper is interesting, well written and gives comprehensive literature review in the area of various diseases such as osteomyelitis, malignant bone tumors, and soft tissue sarcomas localized in this region.
However, final diagnosis was made through surgical biopsy and histopathological and immunohistochemical examination.
Please provide photos or figures of histopathological and immunohistochemical findings.
Best regards,
Prof.dr D. Radojkovic
Author Response
Reviewer B's comments #1> In the paper titled: "Diagnostic Challenge in Rapidly Growing Langerhans Cell Histiocytosis with Aneurysmal Bone Cyst in the Maxilla: A Case Report", the aim was to evaluate numerous differential diagnostic possibilities, according to the radiological, CT and MRI features.The paper is interesting, well written and gives comprehensive literature review in the area of various diseases such as osteomyelitis, malignant bone tumors, and soft tissue sarcomas localized in this region.
However, final diagnosis was made through surgical biopsy and histopathological and immunohistochemical examination.
Please provide photos or figures of histopathological and immunohistochemical findings.
Reply: Thank you very much for your comments. We add the figure 2. (A) Biopsy from the right periorbital mass shows abundant Langerhans cells with scattered eosinophils and multinucleated giant cells (hematoxylin and eosin, magnification x 400). (B) CD1 immunostaining demonstrates membranous staining pattern with perinuclear dot.
Reviewer 3 Report
The authors present a case of rapidly growing mass revealing LCH in a child with a favorable outcome with Vinblatsin/prednisone regimen.
In my opinion, authors must insist in the abstract on the pattern found in tissue biopsy in typical LCH and the frequency of mutations in the MAP-kinase signaling pathway genes.
Thus, they need to provide information and images about tissue biopsy and BRAFV600E status and exclude differential diagnosis on tissue biopsy. BRAFV600 E status is of interest as it correlates with aggressive and relapsing diseases. ( DOI: 10.1200/JCO.2015.65.9508)
The authors describe the differential diagnosis of tumor mass in the discussion, which has not been exposed in the case presentation. They need to provide a deeper analysis of tissue biopsy.
Line 89: the origins of LCH cells are not correct. You might refer to doi: 10.1182/blood.2019000934.
Author Response
Reviewer C's comments #1> The authors present a case of rapidly growing mass revealing LCH in a child with a favorable outcome with Vinblatsin/prednisone regimen.
In my opinion, authors must insist in the abstract on the pattern found in tissue biopsy in typical LCH and the frequency of mutations in the MAP-kinase signaling pathway genes.
Thus, they need to provide information and images about tissue biopsy and BRAFV600E status and exclude differential diagnosis on tissue biopsy. BRAFV600 E status is of interest as it correlates with aggressive and relapsing diseases. ( DOI: 10.1200/JCO.2015.65.9508)
The authors describe the differential diagnosis of tumor mass in the discussion, which has not been exposed in the case presentation. They need to provide a deeper analysis of tissue biopsy.
Reply: Thank you for your comments. First, we add the figure 2. Biopsy from the right periorbital mass shows clusters of histiocytic cells featuring moderate amounts of eosinophilic cytoplasm and reniform nuclei associated with a few eosinophils (hematoxylin and eosin, magnification x 400). These clusters of histiocytic cells show reactivity for CD1a and S-100 and negative for H3, 3G34W. BRAFV600E mutation is not detected.
Furthermore, we add the description about BRAFV600E.
BRAF is considered one of the most common and well-known mutated kinases in human center, which driver mutations in several tumors including colorectal cancer, non-small cell lung cancer, thyroid cancer and melanoma [12, 13]. Its V600E mutation accounts for more than 90% of BRAF-activating mutations [14]. Previous studies have shown that BRAF V600E mutations are present in the average of 47% of LCH cases (from 0 to 64%) [15]. Feng et al. reported that patients from East Asia showed a lower frequency than other ethnic groups, regardless of age [13]. BRAF V600E mutation is not detected in our case. BRAF mutation in LCH cases significantly elevates phosphor-extracellular signal regulated kinase expression which suggested the activation of the mitogen-activated protein kinase pathway, suppressing cell migration and augments cell survival [16]. The clinical significance of BRAF V600E in LCH is still contradictory, because of low incidence and high misdiagnosis rate.
Findings from the present report suggest that in case of a rapidly growing bone tumor with or without an invasive pattern or aneurysmal bone cyst on imaging findings, LCH should be suspected based on the patient’s age and imaging findings. BRAF V600E mutation is helpful for diagnosis and for treatment in LCH patients, we should remember that ethnic background appears to influence the risk of LCH development.
Ref>
- Davies H, Bignell GR, Cox C et al. Mutations of the BRAF gene in human cancer. Nature 2002; 417: 949-954.
- Feng S, Han L, Yue M et al. Frequency detection of BRAF V600E mutation in a cohort of pediatric langerhans cell histiocytosis patients by next-generation sequencing. Orphanet J Rare Dis 2021; 16: 272.
- Thacker NH, Abla O. Pediatric Langerhans cell histiocytosis: state of the science and future directions. Clin Adv Hematol Oncol 2019; 17: 122-131.
- Berres ML, Lim KP, Peters T et al. BRAF-V600E expression in precursor versus differentiated dendritic cells defines clinically distinct LCH risk groups. J Exp Med 2014; 211: 669-683.
- Hogstad B, Berres ML, Chakraborty R et al. RAF/MEK/extracellular signal-related kinase pathway suppresses dendritic cell migration and traps dendritic cells in Langerhans cell histiocytosis lesions. J Exp Med 2018; 215: 319-336.
Reviewer C's comments #2> Line 89: the origins of LCH cells are not correct. You might refer to doi: 10.1182/blood.2019000934.
Reply: Thank you for your comments. We delete the line 89.

Round 2
Reviewer 3 Report
The new version of the paper shows significant improvement in the case presentation, especially on the histological description.
However, a few points might be improved.
In the abstract, the origin of cells is not correct. There are no bone marrow Langerhans cells but CD14+ monocytes bearing BRAF mutation for most ( Durham; Blood 2014).
Furthermore, did the authors perform a phospho-Erk search on biopsy or full NGS?
Author Response
Comment #1>In the abstract, the origin of cells is not correct. There are no bone marrow Langerhans cells but CD14+ monocytes bearing BRAF mutation for most ( Durham; Blood 2014).
--> Replay: Thank you for your comments, We modified the abstract as follows:
Langerhans cell histiocytosis (LCH) is a rare neoplastic disorder characterized by the clonal proliferation of CD1a + / CD 207 + dendritic cells, whose features are similar to those of epidermal Langerhans cells. hematologic disorder characterized by abnormal proliferation of bone marrow-derived Langerhans cells and mature eosinophils.
Comment #2> Furthermore, did the authors perform a phospho-Erk search on biopsy or full NGS?
--> Replay: We also considered ERK search ahd full NGS because the BRAF mutation in LCH lesions signifcantly elevates phospho-extracellular signal-regulated kinase (ERK) expression, suggesting the activation of the mitogen-activated protein kinase (MAPK) pathway. However in East Asian patients, the positive rate of NGS was reported less than 50%, so full NGS study was not done. We do not perform all tests in consideration of cost compared to the benefit that the patient can obtain. And because the patient responded well to the initial treatment, the examination was not performed considering the cost aspect. However, if a difficult case arises later, I would like to conduct additional tests.
REF) #Zeng K, Ohshima K, Liu Y, Zhang W, Wang L, Fan L, et al. BRAFV600E and MAP2K1 mutations in Langerhans cell histiocytosis occur predominantly in children. Hematol Oncol. 2016;35(4):845–51.
#Feng S, Han L, Yue M, Zhong D, Cao J, Guo Y, Sun Y, Zhang H, Cao Z, Cui X et al: Frequency detection of BRAF V600E mutation in a cohort of pediatric langerhans cell histiocytosis patients by next-generation sequencing. Orphanet J Rare Dis 2021, 16(1):272.